# Black-Box Differential Privacy for Interactive ML

**Haim Kaplan**[*]    **Yishay Mansour**[†]    **Shay Moran**[‡]    **Kobbi Nissim**[§]    **Uri Stemmer**[¶]

## Abstract

In this work we revisit an interactive variant of joint differential privacy, recently introduced by Naor et al. [2023], and generalize it towards handling online processes in which existing privacy definitions seem too restrictive. We study basic properties of this definition and demonstrate that it satisfies (suitable variants) of group privacy, composition, and post processing.

In order to demonstrate the advantages of this privacy definition compared to traditional forms of differential privacy, we consider the basic setting of online classification. We show that any (possibly non-private) learning rule can be *effectively* transformed to a private learning rule with only a polynomial overhead in the mistake bound. This demonstrates a stark difference with traditional forms of differential privacy, such as the one studied by Golowich and Livni [2021], where only a double exponential overhead in the mistake bound is known (via an information theoretic upper bound).

## 1 Introduction

In this work we study privacy of interactive machine learning processes. As a motivating story, consider a chatbot that continuously improves itself by learning from the conversations it conducts with users. As these conversations might contain sensitive information, we would like to provide privacy guarantees to the users, in the sense that the content of their conversations with the chatbot would not leak. This setting fleshes out the following two requirements.

(1) Clearly, the answers given by the chatbot to user $u_i$ must depend on the queries made by user $u_i$. For example, the chatbot should provide different answers when asked by user $u_i$ for the weather forecast in Antarctica, and when asked by $u_i$ for a pasta recipe.

   This is in contrast to the plain formulation of differential privacy, where it is required that *all* of the mechanism outputs would be (almost) independent of any single user input. Therefore, the privacy requirement we are aiming for is that the conversation of user $u_i$ will remain "hidden" from *other* users, and would not leak through the *other* users' interactions with the chatbot. Moreover, this should remain true even if a "privacy attacker" (aiming to extract information about the conversation user $u_i$ had) conducts *many* different conversations with the chatbot.

(2) The interaction with the chatbot is, by design, *interactive* and *adaptive*, as it aims to conduct dialogues with the users. This allows the privacy attacker (mentioned above) to choose its queries to the chatbot *adaptively*. Privacy, hence, needs to be preserved even in the presence of adaptive attackers.

While each of these two requirements was studied in isolation, to the best of our knowledge, they have not been (explicitly) unified into a combined privacy framework. Requirement (1) was formalized

---

[*]Tel Aviv University and Google Research. `haimk@tau.ac.il`.

[†]Tel Aviv University and Google Research. `mansour.yishay@gmail.com`.

[‡]Technion and Google Research. `smoran@technion.ac.il`.

[§]Georgetown University. `kobbi.nissim@georgetown.edu`.

[¶]Tel Aviv University and Google Research. `u@uri.co.il`.

37th Conference on Neural Information Processing Systems (NeurIPS 2023).

by Kearns et al. [2015] as *joint differential privacy (JDP)*. It provides privacy against *non-adaptive* attackers. Intuitively, in the chatbot example, JDP aims to hide the conversation of user $u_i$ from any privacy attacker that *chooses in advance* all the queries it poses to the chatbot. This is unsatisfactory since the adaptive nature of this process invites adaptive attackers.

Requirement (2) was studied in many different settings, but to the best of our knowledge, only w.r.t. the plain formulation of DP, where the (adaptive) privacy attacker sees *all* of the outputs of the mechanism. Works in this vein include [Dwork et al., 2009, Chan et al., 2010, Hardt and Rothblum, 2010, Dwork et al., 2010b, Bun et al., 2017, Kaplan et al., 2021, Jain et al., 2021]. In the chatbot example, plain DP would require, in particular, that even the messages sent from the chatbot to user $u_i$ reveals (almost) no information about $u_i$. In theory, this could be obtained by making sure that the *entire chatbot model* is computed in a privacy preserving manner, such that even its full description leaks almost no information about any single user. Then, when user $u_i$ comes, we can "simply" share the model with her, and let her query it locally on her device. But this is likely unrealistic with large models involving hundreds of billions of parameters.

In this work we use *challenge differential privacy*, which was recently introduced by Naor et al. [2023] in the context of PAC learning.[6] As discussed below, challenge differential privacy is particularly suitable for addressing interactive and adaptive learning processes, such as the one illustrated above. Challenge DP can be viewed as an interactive variant of JDP, aimed at maintaining privacy against adaptive privacy attackers. Intuitively, in the chatbot example, this definition would guarantee that even an adaptive attacker that controls *all* of the users except for user $u_i$, learns (almost) no information about the conversation user $u_i$ had with the chatbot.

## 1.1 Private Online Classification

We initiate the study of challenge differential privacy in the basic setting of online classification. Let $\mathcal{X}$ be the domain, $\mathcal{Y}$ be the label space, and $\mathcal{Z} = \mathcal{X} \times \mathcal{Y}$ be set of labeled examples. An online learner is a (possibly randomized) mapping $\mathcal{A} : \mathcal{Z}^\star \times \mathcal{X} \to \mathcal{Y}$. That is, it is a mapping that maps a finite sequence $S \in \mathcal{Z}^\star$ (the past examples), and an unlabeled example $x$ (the current query point) to a label $y$, which is denoted by $y = \mathcal{A}(x; S)$.

Let $\mathcal{H} \subseteq \mathcal{Y}^{\mathcal{X}}$ be a hypothesis class. A sequence $S \in \mathcal{Z}^\star$ is said to be realizable by $\mathcal{H}$ if there exists $h \in \mathcal{H}$ such that $h(x_i) = y_i$ for every $(x_i, y_i) \in S$. For a sequence $S = \{(x_t, y_t)\}_{t=1}^T \in \mathcal{Z}^\star$ we write $\mathcal{M}(\mathcal{A}; S)$ for the random variable denoting the number of mistakes $\mathcal{A}$ makes during the execution on $S$. That is $\mathcal{M}(\mathcal{A}; S) = \sum_{t=1}^T 1\{\hat{y}_t \neq y_t\}$, where $\hat{y}_t = \mathcal{A}(x_t; S_{<t})$ is the (randomized) prediction of $\mathcal{A}$ on $x_t$.

**Definition 1.1** (Online Learnability: Realizable Case). *We say that a hypothesis class $\mathcal{H}$ is online learnable if there exists a learning rule $\mathcal{A}$ such that $\mathbb{E}\left[\mathcal{M}(\mathcal{A}; S)\right] = o(T)$ for every sequence $S$ which is realizable by $\mathcal{H}$.*

**Remark 1.2.** *Notice that Definition 1.1 corresponds to an* oblivious *adversary, as it quantifies over the input sequence in advance. This should not be confused with the adversaries considered in the context of* privacy *which are always adaptive in this work. In the non-private setting, focusing on oblivious adversaries does not affect generality in terms of utility. This is less clear when privacy constraints are involved.[7] We emphasize that our results (our mistake bounds) continue to hold even when the realizable sequence is chosen by an adaptive (stateful) adversary, that at every point in time chooses the next input to the algorithm based on all of the previous outputs of the algorithm.*

A classical result due to Littlestone [1988] characterizes online learnability (without privacy constraints) in terms of the Littlestone dimension. The latter is a combinatorial parameter of $\mathcal{H}$ which was named after Littlestone by Ben-David et al. [2009].

In particular, Littlestone's characterization implies the following dichotomy: if $\mathcal{H}$ has finite Littlestone dimension $d$ then there exists a (deterministic) learning rule which makes at most $d$ mistakes on

---

[6]The privacy notion we study, Challenge DP, is a generalization of the privacy notion presented by Naor et al. [2023]. They focused on a special case in which there is no interaction with individual users. In the chatbot example, this corresponds to having each user submitting only a single query to the chatbot without conducting an adaptive dialog with it. In addition, we analyze useful properties of Challenge DP, such as composition, post-processing, and group-privacy. These properties were not studied by Naor et al. [2023].

[7]In particular, Golowich and Livni [2021] studied both oblivious and adaptive adversaries, and obtained very different results in these two cases.

every realizable input sequence. In the complementing case, when the Littlestone dimension of $\mathcal{H}$ is infinite, for every learning rule $\mathcal{A}$ and every $T \in \mathbb{N}$ there exists a realizable sequence $S$ of length $T$ such that $\mathbb{E}\left[\mathcal{M}(\mathcal{A}; S)\right] \geq T/2$. In other words, as a function of $T$, the optimal mistake bound is either uniformly bounded by the Littlestone dimension, or it is $\geq T/2$. Because of this dichotomy, in some places online learnability is defined with respect to a uniform bound on the number of mistakes (and not just a sublinear one as in the above definition). In this work we follow the more general definition.

We investigate the following questions: *Can every online learnable class be learned by an algorithm which satisfies challenge differential privacy? What is the optimal mistake bound attainable by private learners?*

Our main result in this part provides an affirmative answer to the first question. We show that for any class $\mathcal{H}$ with Littlestone dimension $d$ there exists an $(\varepsilon, \delta)$-challenge-DP learning rule which makes at most $\tilde{O}\left(\frac{d^2}{\varepsilon^2} \log^2\left(\frac{1}{\delta}\right) \log^2\left(\frac{T}{\beta}\right)\right)$ mistakes, with probability $1 - \beta$, on every realizable sequence of length $T$. *Remarkably, our proof provides an efficient transformation taking a non-private learner to a private one:* that is, given a black box access to a learning rule $\mathcal{A}$ which makes at most $M$ mistakes in the realizable case, we efficiently construct an $(\varepsilon, \delta)$-challenge-DP learning rule $\mathcal{A}'$ which makes at most $\tilde{O}\left(\frac{M^2}{\varepsilon^2} \log^2\left(\frac{1}{\delta}\right) \log^2\left(\frac{T}{\beta}\right)\right)$ mistakes.

### 1.1.1 Construction overview

We now give a simplified overview of our construction, called POP, which transforms a non-private online learning algorithm into a private one (while maintaining computational efficiency). Let $\mathcal{A}$ be a non-private algorithm, guaranteed to make at most $d$ mistakes in the realizable setting. We maintain $k$ copies of $\mathcal{A}$. Informally, in every round $i \in [T]$ we do the following:

1. Obtain an input point $x_i$.
2. Give $x_i$ to each of the $k$ copies of $\mathcal{A}$ to obtain predicted labels $\hat{y}_{i,1}, \ldots, \hat{y}_{i,k}$.
3. Output a "privacy preserving" aggregation $\hat{y}_i$ of $\{\hat{y}_{i,1}, \ldots, \hat{y}_{i,k}\}$, which is some variant of noisy majority. This step will only satisfy challenge-DP, rather than (standard) DP.
4. Obtain the "true" label $y_i$.
5. Let $\ell \in [k]$ be chosen at random.
6. Rewind all copies of algorithm $\mathcal{A}$ except for the $\ell$th copy, so that they "forget" ever seeing $x_i$.
7. Give the true label $y_i$ to the $\ell$th copy of $\mathcal{A}$.

As we aggregate the predictions given by the copies of $\mathcal{A}$ using (noisy) majority, we know that if the algorithm errs then at least a constant fraction of the copies of $\mathcal{A}$ err. As we feed the true label $y_i$ to a random copy, with constant probability, the copy which we do not rewind incurs a mistake at this moment. That is, whenever we make a mistake then with constant probability one of the copies we maintain incurs a mistake. This can happen at most $\approx k \cdot d$ times, since we have $k$ copies and each of them makes at most $d$ mistakes. This allows us to bound the number of mistakes made by our algorithm (w.h.p.). The privacy analysis is more involved. Intuitively, by rewinding all of the copies of $\mathcal{A}$ (except one) in every round, we make sure that a single user can affect the inner state of at most one of the copies. This allows us to efficiently aggregate the predictions given by the copies in a privacy preserving manner. The subtle point is that the prediction we release in time $i$ *does* require querying *all* the experts on the current example $x_i$ (before rewinding them). Nevertheless, we show that this algorithm is private.

### 1.1.2 Comparison with Golowich and Livni [2021]

The closest prior work to this manuscript is by Golowich and Livni who also studied the problem of private online classification, but under a more restrictive notion of privacy than challenge-DP. In particular their definition requires that the sequence of predictors which the learner uses to predict in each round does not compromise privacy. In other words, it is as if at each round the learner publishes the entire truth-table of its predictor, rather than just its current prediction. This might be too prohibitive in certain applications such as the chatbot example illustrated above. Golowich and Livni show that even with respect to their more restrictive notion of privacy it is possible to online

learn every Littlestone class. However, their mistake bound is doubly exponential in the Littlestone dimension (whereas ours is quadratic), and their construction requires more elaborate access to the non-private learner. In particular, it is not clear whether their construction can be implemented efficiently (while our construction is efficient).

## 1.2 Additional Related Work

Several works studied the related problem of *private learning from expert advice* [Dwork et al., 2010a, Jain et al., 2012, Thakurta and Smith, 2013, Dwork and Roth, 2014, Jain and Thakurta, 2014, Agarwal and Singh, 2017, Asi et al., 2022]. These works study a variant of the experts problem in which the learning algorithm has access to $k$ *experts*; on every time step the learning algorithm chooses one of the experts to follow, and then observes the *loss* of each expert. The goal of the learning algorithm is that its accumulated loss will be competitive with the loss of the *best expert in hindsight*. In this setting the private data is the sequence of losses observed throughout the execution, and the privacy requirement is that the sequence of experts chosen by the algorithm should not compromise the privacy of the sequence of losses.[8] When applying these results to our context, the set of experts is the set of hypotheses in the class $\mathcal{H}$, which means that the outcome of the learner (on every time step) is a complete model (i.e., a hypothesis). That is, in our context, applying prior works on private prediction from expert advice would result in a privacy definition similar to that of Golowich and Livni [2021] that accounts (in the privacy analysis) for releasing complete models, rather than just the predictions, which is significantly more restrictive.

There were a few works that studied private learning in online settings under the constraint of JDP. For example, Shariff and Sheffet [2018] studied the stochastic contextual linear bandits problem under JDP. Here, in every round $t$ the learner receives a *context* $c_t$, then it selects an *action* $a_t$ (from a fixed set of actions), and finaly it receives a reward $y_t$ which depends on $(c_t, a_t)$ in a linear way. The learner's objective is to maximize cumulative reward. The (non-adaptive) definition of JDP means that action $a_t$ is revealed only to user $u_t$. Furthermore, it guarantees that the inputs of user $u_t$ (specifically the context $c_t$ and the reward $y_t$) do not leak to the other users via the actions they are given, provided that all these other users *fix their data in advance*. This non-adaptive privacy notion fits the stochastic setting of Shariff and Sheffet [2018], but (we believe) is less suited for adversarial processes like the ones we consider in this work. We also note that the algorithm of Shariff and Sheffet [2018] in fact satisfies the more restrictive privacy definition which applies to the sequence of predictors (rather than the sequence of predictions), similarly to the that of Golowich and Livni [2021].

## 2 Preliminaries

**Notation.** Two datasets $S$ and $S'$ are called *neighboring* if one is obtained from the other by adding or deleting one element, e.g., $S' = S \cup \{x'\}$. For two random variables $Y, Z$ we write $X \approx_{(\varepsilon,\delta)} Y$ to mean that for every event $F$ it holds that $\Pr[X \in F] \leq e^{\varepsilon} \cdot \Pr[Y \in F] + \delta$, and $\Pr[Y \in F] \leq e^{\varepsilon} \cdot \Pr[X \in F] + \delta$. Throughout the paper we assume that the privacy parameter $\varepsilon$ satisfies $\varepsilon = O(1)$, but our analyses trivially extend to larger values of epsilon.

The standard definition of differential privacy is,

**Definition 2.1** ([Dwork et al., 2006])**.** *Let $\mathcal{M}$ be a randomized algorithm that operates on datasets. Algorithm $\mathcal{M}$ is $(\varepsilon, \delta)$-differentially private (DP) if for any two neighboring datasets $S, S'$ we have $\mathcal{M}(S) \approx_{(\varepsilon,\delta)} \mathcal{M}(S')$.*

**The Laplace mechanism.** The most basic constructions of differentially private algorithms are via the Laplace mechanism as follows.

**Definition 2.2.** *A random variable has probability distribution $\mathrm{Lap}(\gamma)$ if its probability density function is $f(x) = \frac{1}{2\gamma}\exp(-|x|/\gamma)$, where $x \in \mathbb{R}$.*

**Definition 2.3** (Sensitivity)**.** *A function $f$ that maps datasets to the reals has* sensitivity $\Delta$ *if for every two neighboring datasets $S$ and $S'$ it holds that $|f(S) - f(S')| \leq \Delta$.*

---

[8]Asi et al. [2022] study a more general framework of adaptive privacy in which the private data is an auxiliary sequence $(z_1, \ldots, z_T)$. During the interaction with the learner, these $z_t$'s are used (possibly in an adaptive way) to choose the sequence of loss functions.

**Theorem 2.4** (The Laplace Mechanism [Dwork et al., 2006]). *Let $f$ be a function that maps datasets to the reals with sensitivity $\Delta$. The mechanism $\mathcal{A}$ that on input $S$ adds noise with distribution $\mathrm{Lap}(\frac{\Delta}{\varepsilon})$ to the output of $f(S)$ preserves $(\varepsilon, 0)$-differential privacy.*

**Joint differential privacy.**    The standard definition of differential privacy (Definition 2.1) captures a setting in which the entire output of the computation may be publicly released without compromising privacy. While this is a very desirable requirement, it is sometimes too restrictive. Indeed, Kearns et al. [2015] considered a relaxed setting in which we aim to analyze a dataset $S = (x_1, \ldots, x_n)$, where every $x_i$ represents the information of user $i$, and to obtain a vector of outcomes $(y_1, \ldots, y_n)$. This vector, however, is not made public. Instead, every user $i$ only receives its "corresponding outcome" $y_i$. This setting potentially allows the outcome $y_i$ to strongly depend on the the input $x_i$, without compromising the privacy of the $i$th user from the view point of the other users.

**Definition 2.5** ([Kearns et al., 2015]). *Let $\mathcal{M} : X^n \to Y^n$ be a randomized algorithm that takes a dataset $S \in X^n$ and outputs a vector $\vec{y} \in Y^n$. Algorithm $\mathcal{M}$ satisfies $(\varepsilon, \delta)$-joint differential privacy (JDP) if for every $i \in [n]$ and every two datasets $S, S' \in X^n$ differing only on their $i$th point it holds that $\mathcal{M}(S)_{-i} \approx_{(\varepsilon, \delta)} \mathcal{M}(S')_{-i}$. Here $\mathcal{M}(S)_{-i}$ denotes the (random) vector of length $n-1$ obtained by running $(y_1, \ldots, y_n) \leftarrow \mathcal{M}(S)$ and returning $(y_1, \ldots, y_{i-1}, y_{i+1}, \ldots, y_n)$.*

In words, consider an algorithm $\mathcal{M}$ that operates on the data of $n$ individuals and outputs $n$ outcomes $y_1, \ldots, y_n$. This algorithm is JDP if changing only the $i$th input point $x_i$ has almost no affect on the outcome distribution of the *other* outputs (but the outcome distribution of $y_i$ is allowed to strongly depend on $x_i$). Kearns et al. [2015] showed that this fits a wide range of problems in economic environments.

**Example 2.6** ([Nahmias et al., 2019]). *Suppose that a city water corporation is interested in promoting water conservation. To do so, the corporation decided to send each household a customized report indicating whether their water consumption is above or below the median consumption in the neighborhood. Of course, this must be done in a way that protects the privacy of the neighbors. One way to tackle this would be to compute a privacy preserving estimation $z$ for the median consumption (satisfying Definition 2.1). Then, in each report, we could safely indicate whether the household's water consumption is bigger or smaller than $z$. While this solution is natural and intuitive, it turns out to be sub-optimal: We can obtain better utility by designing a JDP algorithm that directly computes a different outcome for each user ("above" or "below"), which is what we really aimed for, without going through a private median computation.*

**Algorithm `AboveThreshold`.**    Consider a large number of low sensitivity functions $f_1, f_2, \ldots, f_T$ which are given (one by one) to a data curator (holding a dataset $S$). Algorithm `AboveThreshold` allows for privately identifying the queries $f_i$ whose value $f_i(S)$ is (roughly) greater than some threshold $t$.

---

**Algorithm 1 `AboveThreshold` [Dwork et al., 2009, Hardt and Rothblum, 2010]**

**Input:** Dataset $S \in X^*$, privacy parameters $\varepsilon, \delta$, threshold $t$, number of positive reports $r$, and an adaptively chosen stream of queries $f_i : X^* \to \mathbb{R}$ with sensitivity $\Delta$

  1. Denote $\gamma = O\left(\frac{\Delta}{\varepsilon} \sqrt{r} \ln(\frac{r}{\delta})\right)$

  2. In each round $i$, when receiving a query $f_i \in Q$, do the following:

    (a) Let $\hat{f}_i \leftarrow f_i(S) + \mathrm{Lap}(\gamma)$

    (b) If $\hat{f}_i \geq t$, then let $\sigma_i = 1$ and otherwise let $\sigma_i = 0$

    (c) Output $\sigma_i$

    (d) If $\sum_{j=1}^{i} \sigma_j \geq r$ then HALT

---

Even though the number of possible rounds is unbounded, algorithm `AboveThreshold` preserves differential privacy. Note, however, that `AboveThreshold` is an *interactive* mechanism, while the standard definition of differential privacy (Definition 2.1) is stated for *non-interactive* mechanisms, that process their input dataset, release an output, and halt. The adaptation of DP to such interactive settings is done via a *game* between the (interactive) mechanism and an *adversary* that specifies the inputs to the mechanism and observes its outputs. Intuitively, the privacy requirement is that the view

of the adversary at the end of the execution should be differentially private w.r.t. the inputs given to the mechanism. Formally,

**Definition 2.7** (DP under adaptive queries [Dwork et al., 2006, Bun et al., 2017]). *Let $\mathcal{M}$ be a mechanism that takes an input dataset and answers a sequence of adaptively chosen queries (specified by an adversary $\mathcal{B}$ and chosen from some family $Q$ of possible queries). Mechanism $\mathcal{M}$ is $(\varepsilon, \delta)$-differentially private if for every adversary $\mathcal{B}$ we have that $\mathtt{AdaptiveQuery}_{\mathcal{M}, \mathcal{B}, Q}$ (defined below) is $(\varepsilon, \delta)$-differentially private (w.r.t. its input bit $b$).*

---

**Algorithm 2** $\mathtt{AdaptiveQuery}_{\mathcal{M}, \mathcal{B}, Q}$ **[Bun et al., 2017]**

---

**Input:** A bit $b \in \{0, 1\}$. (The bit $b$ is unknown to $\mathcal{M}$ and $\mathcal{B}$.)
1. The adversary $\mathcal{B}$ chooses two neighboring datasets $S_0$ and $S_1$.
2. The dataset $S_b$ is given to the mechanism $\mathcal{M}$.
3. For $i = 1, 2, \ldots$
    (a) The adversary $\mathcal{B}$ chooses a query $q_i \in Q$.
    (b) The mechanism $\mathcal{M}$ is given $q_i$ and returns $a_i$.
    (c) $a_i$ is given to $\mathcal{B}$.
4. When $\mathcal{M}$ or $\mathcal{B}$ halts, output $\mathcal{B}$'s view of the interaction.

---

**Theorem 2.8** ([Dwork et al., 2009, Hardt and Rothblum, 2010, Kaplan et al., 2021]). *Algorithm $\mathtt{AboveThreshold}$ is $(\varepsilon, \delta)$-differentially private.*

**A private counter.** In the setting of algorithm $\mathtt{AboveThreshold}$, the dataset is fixed in the beginning of the execution, and the queries arrive sequentially one by one. Dwork et al. [2010a] and Chan et al. [2010] considered a different setting, in which the *data* arrives sequentially. In particular, they considered the *counter* problem where in every time step $i \in [T]$ we obtain an input bit $x_i \in \{0, 1\}$ (representing the data of user $i$) and must immediately respond with an approximation for the current sum of the bits. That is, at time $i$ we wish to release an approximation for $x_1 + x_2 + \cdots + x_i$.

Similarly to our previous discussion, this is an *interactive* setting, and privacy is defined via a *game* between a mechanism $\mathcal{M}$ and an adversary $\mathcal{B}$ that adaptively determines the inputs for the mechanism.

**Definition 2.9** (DP under adaptive inputs [Dwork et al., 2006, 2010a, Chan et al., 2010, Kaplan et al., 2021, Jain et al., 2021]). *Let $\mathcal{M}$ be a mechanism that in every round $i$ obtains an input point $x_i$ (representing the information of user $i$) and outputs a response $a_i$. Mechanism $\mathcal{M}$ is $(\varepsilon, \delta)$-differentially private if for every adversary $\mathcal{B}$ we have that $\mathtt{AdaptiveInput}_{\mathcal{M}, \mathcal{B}}$ (defined below) is $(\varepsilon, \delta)$-differentially private (w.r.t. its input bit $b$).*

---

**Algorithm 3** $\mathtt{AdaptiveInput}_{\mathcal{M}, \mathcal{B}}$ **[Jain et al., 2021]**

---

**Input:** A bit $b \in \{0, 1\}$. (The bit $b$ is unknown to $\mathcal{M}$ and $\mathcal{B}$.)
1. For $i = 1, 2, \ldots$
    (a) The adversary $\mathcal{B}$ outputs a bit $c_i \in \{0, 1\}$, under the restriction that $\sum_{j=1}^{i} c_j \leq 1$.
        % The round $i$ in which $c_i = 1$ is called the *challenge* round. Note that there could be at most one challenge round throughout the game.
    (b) The adversary $\mathcal{B}$ chooses two input points $x_{i,0}$ and $x_{i,1}$, under the restriction that if $c_i = 0$ then $x_{i,0} = x_{i,1}$.
    (c) Algorithm $\mathcal{M}$ obtains $x_{i,b}$ and outputs $a_i$.
    (d) $a_i$ is given to $\mathcal{B}$.
2. When $\mathcal{M}$ or $\mathcal{B}$ halts, output $\mathcal{B}$'s view of the interaction.

---

**Theorem 2.10** (Private counter [Dwork et al., 2010a, Chan et al., 2010, Jain et al., 2021]). *There exists an $(\varepsilon, 0)$-differentially private mechanism $\mathcal{M}$ (as in Definition 2.9) that in each round $i \in [T]$ obtains an input bit $x_i \in \{0, 1\}$ and outputs a response $a_i \in \mathbb{N}$ with the following properties. Let $s$ denote the random coins of $\mathcal{M}$. Then there exists an event $E$ such that: (1) $\Pr[s \in E] \geq 1 - \beta$, and (2) Conditioned on every $s \in E$, for every input sequence $(x_1, \ldots, x_T)$, the answers $(a_1, \ldots, a_T)$ satisfy $\left| a_i - \sum_{j=1}^{i} x_i \right| \leq O\left( \frac{1}{\varepsilon} \log(T) \log\left( \frac{T}{\beta} \right) \right)$.*

# 3 Challenge Differential Privacy

We now introduce the privacy definition we consider in this work is. Intuitively, the requirement is that even an adaptive adversary controlling all of the users except Alice, cannot learn much information about the interaction Alice had with the algorithm.

**Definition 3.1** (Extension of Naor et al. [2023]). *Consider an algorithm $\mathcal{M}$ that, in each round $i \in [T]$ obtains an input point $x_i$, outputs a "predicted" label $\hat{y}_i$, and obtains a "true" label $y_i$. We say that algorithm $\mathcal{M}$ is $(\varepsilon, \delta)$-challenge differentially private if for any adversary $\mathcal{B}$ we have that $\texttt{OnlineGame}_{\mathcal{M},\mathcal{B},T}$, defined below, is $(\varepsilon, \delta)$-differentially private (w.r.t. its input bit b).*

---

**Algorithm 4** $\texttt{OnlineGame}_{\mathcal{M},\mathcal{B},T,g}$

**Setting:** $T \in \mathbb{N}$ denotes the number of rounds and $g \in \mathbb{N}$ is a "group privacy" parameter. If not explicitly stated we assume that $g = 1$. $\mathcal{M}$ is an online algorithm and $\mathcal{B}$ is an adversary that determines the inputs adaptively.

**Input of the game:** A bit $b \in \{0, 1\}$. (The bit $b$ is unknown to $\mathcal{M}$ and $\mathcal{B}$.)
1. For $i = 1, 2, \ldots, T$
   (a) The adversary $\mathcal{B}$ outputs a bit $c_i \in \{0, 1\}$, under the restriction that $\sum_{j=1}^{i} c_j \leq g$.
      % We interpret rounds $i$ in which $c_i = 1$ as *challenge* rounds. Note that there could be at most $g$ challenge rounds throughout the game.
   (b) The adversary $\mathcal{B}$ chooses two labeled inputs $(x_{i,0}, y_{i,0})$ and $(x_{i,1}, y_{i,1})$, under the restriction that if $c_i = 0$ then $(x_{i,0}, y_{i,0}) = (x_{i,1}, y_{i,1})$.
   (c) Algorithm $\mathcal{M}$ obtains $x_{i,b}$, then outputs $\hat{y}_i$, and then obtains $y_{i,b}$.
   (d) If $c_i = 0$ then set $\tilde{y}_i = \hat{y}_i$. Otherwise set $\tilde{y}_i = \bot$.
   (e) The adversary $\mathcal{B}$ obtains $\tilde{y}_i$.
      % Note that the adversary $\mathcal{B}$ does not get to see the outputs of $\mathcal{M}$ in challenge rounds.
2. Output $\mathcal{B}$'s view of the game, that is $\tilde{y}_1, \ldots, \tilde{y}_T$ and the internal randomness of $\mathcal{B}$.
   % Note that from this we can reconstruct all the inputs specified by $\mathcal{B}$ throughout the game.

---

**Remark 3.2.** *Naor et al. [2023] studied a special case of this definition, suitable to their stochastic setting. More specifically, they considered a setting where the algorithm initially gets a dataset containing labeled examples. Then, on every time step, a new user arrives and submits its unlabeled example to the algorithm, and the algorithm responds with a predicted label. We extend the definition to capture settings in which every user interacts with the algorithm (rather than just submitting its input). In the concrete application we consider (online learning) this corresponds to the user submitting its input, then obtaining the predicted label, and then submitting the "true" label. Our generalized definition (Section A) captures this as a special case and allows for arbitrary interactions with each user.*

**Composition and post-processing.** Composition and post-processing for challenge-DP follows immediately from their analogues for (standard) DP. Formally, composition is defined via the following game, called $\texttt{CompositionGame}$, in which a "meta adversary" $\mathcal{B}^*$ is trying to guess an unknown bit $b \in \{0, 1\}$. The meta adversary $\mathcal{B}^*$ is allowed to (adaptively) invoke $k$ executions of the game specified in Algorithm 4, where all of these $k$ executions are done with the same (unknown) bit $b$. See Algorithm 5.

---

**Algorithm 5** $\texttt{CompositionGame}_{\mathcal{B}^*,m,\varepsilon,\delta}$

**Input of the game:** A bit $b \in \{0, 1\}$. (The bit $b$ is unknown to $\mathcal{B}^*$.)
1. For $\ell = 1, 2, \ldots, m$
   (a) The adversary $\mathcal{B}^*$ outputs an $(\varepsilon, \delta)$-challenge-DP algorithm $\mathcal{M}_\ell$, an adversary $\mathcal{B}_\ell$, and an integer $T_\ell$.
   (b) The adversary $\mathcal{B}^*$ obtains the outcome of $\texttt{OnlineGame}_{\mathcal{M}_\ell,\mathcal{B}_\ell,T_\ell}(b)$.
2. Output $\mathcal{B}^*$'s view of the game (its internal randomness and all of the outcomes of $\texttt{OnlineGame}$ it obtained throughout the execution).

---

The following theorem follows immediately from standard composition theorems for differential privacy [Dwork et al., 2010b].

**Theorem 3.3** (special case of [Dwork et al., 2010b]). *For every $\mathcal{B}^*$, every $m \in \mathbb{N}$ and every $\varepsilon, \delta, \delta' \geq 0$ it holds that $\texttt{CompositionGame}_{\mathcal{B}^*, m, \varepsilon, \delta}$ is $(\varepsilon', m\delta + \delta')$-differentially private (w.r.t. the input bit $b$) for $\varepsilon' = \sqrt{2m \ln(1/\delta')}\varepsilon + m\varepsilon(e^\varepsilon - 1)$.*

**Group privacy.** We show that challenge-DP is closed under group privacy. This is more subtle than the composition argument. In fact, we first need to *define* what do we mean by "group privacy" in the context of challenge-DP, which we do using the parameter $g$ in algorithm $\texttt{OnlineGame}$. Recall that throughout the execution of $\texttt{OnlineGame}$, the adversary is allowed $g$ challenge rounds. We show that if an algorithm satisfies challenge-DP when the adversary is allowed only a single challenge round, then it also satisfies challenge-DP (with related privacy parameters) when the adversary is allowed $g > 1$ challenge rounds. This is captured by the following theorem; see Appendix B for the proof.

**Theorem 3.4.** *Let $\mathcal{M}$ be an algorithm that in each round $i \in [T]$ obtains an input point $x_i$, outputs a "predicted" label $\hat{y}_i$, and obtains a "true" label $y_i$. If $\mathcal{M}$ is $(\varepsilon, \delta)$-challenge-DP then for every $g \in \mathbb{N}$ and every adversary $\mathcal{B}$ (posing at most $g$ challenges) we have that $\texttt{OnlineGame}_{\mathcal{M}, \mathcal{B}, T, g}$ is $(g\varepsilon, g \cdot e^{\varepsilon g} \cdot \delta)$-differentially private.*

## 4 Algorithm `ChallengeAT`

Towards presenting our private online learner, we introduce a variant of algorithm $\texttt{AboveThreshold}$ with additional guarantees, which we call $\texttt{ChallengeAT}$. Recall that $\texttt{AboveThreshold}$ "hides" arbitrary modifications to a single input point. Intuitively, the new variant we present aims to hide both an arbitrary modification to a single input point and an arbitrary modification to a single query throughout the execution. Consider algorithm $\texttt{ChallengeAT}$.

---

**Algorithm 6 `ChallengeAT`**

---

**Input:** Dataset $S \in X^*$, privacy parameters $\varepsilon, \delta$, threshold $t$, number of positive reports $r$, and an adaptively chosen stream of queries $f_i : X^* \to \mathbb{R}$ each with sensitivity $\Delta$

**Tool used:** An $(\varepsilon, 0)$-DP algorithm, $\texttt{PrivateCounter}$, for counting bits under continual observation, guaranteeing error at most $\lambda$ with probability at least $1 - \delta$

1. Instantiate $\texttt{PrivateCounter}$
2. Denote $\gamma = O\left(\frac{\Delta}{\varepsilon}\sqrt{r + \lambda}\ln(\frac{r+\lambda}{\delta})\right)$
3. In each round $i$, when receiving a query $f_i$, do the following:
    (a) Let $\hat{f}_i \leftarrow f_i(S) + \mathrm{Lap}(\gamma)$
    (b) If $\hat{f}_i \geq t$, then let $\sigma_i = 1$ and otherwise let $\sigma_i = 0$
    (c) Output $\sigma_i$
    (d) Feed $\sigma_i$ to $\texttt{PrivateCounter}$ and let $\mathrm{count}_i$ denote its current output
    (e) If $\mathrm{count}_i \geq r$ then HALT

---

**Remark 4.1.** *When we apply $\texttt{ChallengeAT}$, it sets $\lambda = O\left(\frac{1}{\varepsilon}\log(T)\log\left(\frac{T}{\beta}\right)\right)$. Technically, for this it has to know $T$ and $\beta$. To simplify the description this is not explicit in our algorithms.*

The utility guarantees of $\texttt{ChallengeAT}$ are straightforward. The following theorem follows by bounding (w.h.p.) all the noises sampled throughout the execution (when instantiating $\texttt{ChallengeAT}$ with the private counter from Theorem 2.10).

**Theorem 4.2.** *Let $s$ denote the random coins of $\texttt{ChallengeAT}$. Then there exists an event $E$ such that: (1) $\Pr[s \in E] \geq 1 - \beta$, and (2) Conditioned on every $s \in E$, for every input dataset $S$ and every sequence of $T$ queries $(f_1, \ldots, f_T)$ it holds that*

1. *Algorithm $\texttt{ChallengeAT}$ does not halt before the $r$th time in which it outputs $\sigma_i = 1$,*

2. *For every $i$ such that $\sigma_i = 1$ it holds that $f_i(S) \geq t - O\left(\frac{\Delta}{\varepsilon}\sqrt{r + \lambda}\ln(\frac{r+\lambda}{\delta})\log(\frac{T}{\beta})\right)$,*

3. *For every $i$ such that $\sigma_i = 0$ it holds that $f_i(S) \leq t + O\left(\frac{\Delta}{\varepsilon}\sqrt{r + \lambda}\ln(\frac{r+\lambda}{\delta})\log(\frac{T}{\beta})\right)$,*

*where $\lambda = O\left(\frac{1}{\varepsilon}\log(T)\log\left(\frac{T}{\beta}\right)\right)$ is the error of the counter of Theorem 2.10.*

The event $E$ in Theorem 4.2 occurs when all the Laplace noises of the counter and `ChallengeAT` are within a factor of $\log(T/\beta)$ of their expectation. The privacy guarantees of `ChallengeAT` are more involved. They are defined via a game with an adversary $\mathcal{B}$ whose goal is to guess a secret bit $b$. At the beginning of the game, the adversary chooses two neighboring datasets $S_0, S_1$, and `ChallengeAT` is instantiated with $S_b$. Then throughout the game the adversary specifies queries $f_i$ and observes the output of `ChallengeAT` on these queries. At some special round $i^*$, chosen by the adversary, the adversary specifies *two* queries $f_{i^*}^0, f_{i^*}^1$, where only $f_{i^*}^b$ is fed into `ChallengeAT`. In round $i^*$ the adversary does not get to see the answer of `ChallengeAT` on $f_{i^*}^b$ (otherwise it could easily learn the bit $b$ since $f_{i^*}^0, f_{i^*}^1$ may be very different). We show that any such adversary $\mathcal{B}$ has only a small advantage in guessing the bit $b$. The formal details are given in Appendix C.

# 5 Online Classification under Challenge Differential Privacy

We are now ready to present our private online prediction algorithm. Consider algorithm `POP`.

**Theorem 5.1.** *When executed with a learner $\mathcal{A}$ that makes at most $d$ mistakes and with parameters $k = \tilde{O}\left(\frac{d}{\varepsilon^2}\log^2(\frac{1}{\delta})\log^2(\frac{T}{\beta})\right)$ and $r = O\left(dk + \ln\left(\frac{1}{\beta}\right)\right)$, then with probability at least $(1-\beta)$ the number of mistakes made by algorithm `POP` is bounded by $\tilde{O}\left(\frac{d^2}{\varepsilon^2}\log^2(\frac{1}{\delta})\log^2(\frac{T}{\beta})\right)$.*

---

**Algorithm 7** `POP` (Private Online Procedure)

---

**Setting:** $T \in \mathbb{N}$ denotes the number of rounds in the game. $\mathcal{A}$ is a non-private online-algorithm.
**Parameters:** $k$ determines the number of copies of $\mathcal{A}$ we maintain. $r$ determines the number of positive reports we aim to receive from `ChallengeAT`.
1. Instantiate $k$ copies $\mathcal{A}_1, \ldots, \mathcal{A}_k$ of algorithm $\mathcal{A}$
2. Instantiate algorithm `ChallengeAT` on an empty dataset with threshold $t = -k/4$, privacy parameters $\varepsilon, \delta$, number of positive reports $r$, and sensitivity parameter $\Delta = 1$.
3. For $i = 1, 2, \ldots, T$
   (a) Obtain input $x_i$
   (b) Let $\mathcal{A}_1^{\text{temp}}, \ldots, \mathcal{A}_k^{\text{temp}}$ be duplicated copies of $\mathcal{A}_1, \ldots, \mathcal{A}_k$
   (c) Let $\ell_i \in [k]$ be chosen uniformly at random
   (d) Let $\hat{y}_{i,\ell_i} \leftarrow \mathcal{A}_{\ell_i}(x_i)$. For $j \in [k] \setminus \{\ell_i\}$ let $\hat{y}_{i,j} \leftarrow \mathcal{A}_j^{\text{temp}}(x_i)$
   (e) Feed `ChallengeAT` the query $f_i \equiv -\left|\frac{k}{2} - \sum_{j\in[k]}\hat{y}_{i,j}\right|$ and obtain an outcome $\sigma_i$. (If `ChallengeAT` halts then `POP` also halts.)
      % Recall that $\sigma_i = 1$ indicates that $-\left|\frac{k}{2} - \sum_{j\in[k]}\hat{y}_{i,j}\right| \gtrsim -\frac{k}{4}$, meaning that there is "a lot" of disagreement among $\hat{y}_{i,1}, \ldots, \hat{y}_{i,k}$.
   (f) If $\sigma_i = 1$ then sample $\hat{y}_i \in \{0,1\}$ at random. Else let $\hat{y}_i = \text{majority}\{\hat{y}_{i,1}, \ldots, \hat{y}_{i,k}\}$
   (g) Output the bit $\hat{y}_i$ as a prediction, and obtain a "true" label $y_i$
   (h) Feed $y_i$ to $\mathcal{A}_{\ell_i}$
      % Note that $\mathcal{A}_\ell$ is the only copy of $\mathcal{A}$ that changes its state during this iteration

---

Informally, the privacy guarantees of `POP` follow from those of `ChallengeAT`. We give the formal details in Appendix D, obtaining the following theorem.

**Theorem 5.2.** *Algorithm `POP` is $(O(\varepsilon), O(\delta))$-Challenge-DP. That is, For every adversary $\mathcal{B}$ it holds that $\mathit{OnlineGame}_{POP,\mathcal{B}}$ is $(O(\varepsilon), O(\delta))$-DP w.r.t. the bit $b$ (the input of the game).*

We proceed with the utility guarantees of `POP`. See Appendix F for an extension to the agnostic setting.

**Theorem 5.3.** *When executed with a learner $\mathcal{A}$ that makes at most $d$ mistakes and with parameters $k = \tilde{O}\left(\frac{d}{\varepsilon^2}\log^2(\frac{1}{\delta})\log^2(\frac{T}{\beta})\right)$ and $r = O\left(dk + \ln\left(\frac{1}{\beta}\right)\right)$, then with probability at least $(1-\beta)$ the number of mistakes made by algorithm `POP` is bounded by $\tilde{O}\left(\frac{d^2}{\varepsilon^2}\log^2(\frac{1}{\delta})\log^2(\frac{T}{\beta})\right)$.*

*Proof.* By Theorem 4.2, with probability $(1 - \beta)$ over the internal coins of `ChallengeAT`, for every input sequence, its answers are accurate up to error of

$$\text{error}_{\text{CAT}} = O\left(\frac{\Delta}{\varepsilon}\sqrt{r + \lambda}\ln(\frac{r + \lambda}{\delta})\log(\frac{T}{\beta})\right),$$

where in our case, the sensitivity $\Delta$ is 1, and the error of the counter $\lambda$ is at most $O\left(\frac{1}{\varepsilon}\log(T)\log\left(\frac{T}{\delta}\right)\right)$ by Theorem 2.10. We continue with the proof assuming that this event occurs. Furthermore, we set $k = \Omega\left(\text{error}_{\text{CAT}}\right)$, large enough, such that if less than $\frac{1}{5}$ the experts disagree with the other experts, then algorithm `POP` returns the majority vote with probability 1.

Consider the execution of algorithm `POP` and define $1/5$-Err be a random variable that counts the number of time steps in which at least $1/5$th of the experts make an error. That is

$$1/5\text{-Err} = \left|\left\{i \in [T] : \sum_{j \in [k]} \mathbb{1}\{\hat{y}_{i,j} \neq y_i\} > k/5\right\}\right|.$$

We also define the random variable

$$\text{expertAdvance} = |\{i \in [T] : y_i \neq \hat{y}_{i,\ell_i}\}|.$$

That is expertAdvance counts the number of times steps in which the random expert we choose (the $\ell_i$th expert) errs. Note that the $\ell_i$th expert is the expert that gets the "true" label $y_i$ as feedback. As we run $k$ experts, and as each of them is guaranteed to make at most $d$ mistakes, we get that

$$\text{expertAdvance} \leq kd.$$

We now show that with high probability $1/5$-Err is not much larger than expertAdvance. Let $i$ be a time step in which at least $1/5$ fraction of the experts err. As the choice of $\ell_i$ (the expert we update) is random, then with probability at least $\frac{1}{5}$ the chosen expert also errs. It is therefore unlikely that $1/5$-Err is much larger than expertAdvance, which is bounded by $kd$. Specifically, by standard concentration arguments (see Appendix E for the precise version we use) it holds that

$$\Pr\left[1/5\text{-Err} > 18dk + 18 + \ln\left(\frac{1}{\beta}\right)\right] \leq \beta.$$

Note that when at least $1/5$ of the experts disagree with other experts then at least $1/5$ of the experts err. It follows that $1/5$-Err upper bounds the number of times in which algorithm `ChallengeAT` returns an "above threshold" answer. Hence, by setting $r > 18dk + 18 + \ln\left(\frac{1}{\beta}\right)$ we ensure that w.h.p. algorithm `ChallangeAT` does not halt prematurely (and hence `POP` does not either).

Furthermore our algorithm errs either when there is a large disagreement between the experts or when all experts err. It follows that $1/5$-Err also upper bounds the number of times which our algorithm errs.

Overall, by setting $r = O\left(dk + \ln\left(\frac{1}{\beta}\right)\right)$ we ensure that `POP` does not halt prematurely, and by setting $k = O\left(\frac{\Delta}{\varepsilon}\sqrt{r + \lambda}\ln(\frac{r+\lambda}{\delta})\log(\frac{T}{\beta})\right)$ we ensure that `POP` does not err too many times throughout the execution. Combining the requirement on $r$ and on $k$, it suffices to take

$$k = \tilde{O}\left(\frac{d}{\varepsilon^2}\log^2(\frac{1}{\delta})\log^2(\frac{T}{\beta}) + \frac{1}{\varepsilon \cdot d}\log(T)\log\left(\frac{T}{\delta}\right)\right),$$

in which case algorithm `POP` makes at most $\tilde{O}\left(\frac{d^2}{\varepsilon^2}\log^2(\frac{1}{\delta})\log^2(\frac{T}{\beta})\right)$ with high probability.  $\square$

## 6   Conclusion

Our work presents a new privacy model for online classification, together with an *efficiency preserving* transformation from *non-private* online classification, that exhibits a doubly exponential improvement in the error compared to prior works on this topic. We leave open the possibility that such an improvement could also be achieved in the setting of Golowich and Livni [2021], i.e., under the more restrictive notion of privacy where the sequence of predictors does not compromise privacy.

## Acknowledgments and Disclosure of Funding

Haim Kaplan was artially supported by Israel Science Foundation (grant 1595/19), and the Blavatnik Family Foundation.

Yishay Mansour was partially funded from the European Research Council (ERC) under the European Union's Horizon 2020 research and innovation program (grant agreement No. 882396), by the Israel Science Foundation (grant number 993/17), Tel Aviv University Center for AI and Data Science (TAD), and the Yandex Initiative for Machine Learning at Tel Aviv University.

Shay Moran is a Robert J. Shillman Fellow; he acknowledges support by ISF grant 1225/20, by BSF grant 2018385, by an Azrieli Faculty Fellowship, by Israel PBC-VATAT, by the Technion Center for Machine Learning and Intelligent Systems (MLIS), and by the European Union (ERC, GENERALIZATION, 101039692). Views and opinions expressed are however those of the author(s) only and do not necessarily reflect those of the European Union or the European Research Council Executive Agency. Neither the European Union nor the granting authority can be held responsible for them.

Kobbi Nissim was partially funded by NSF grant No. CNS 2001041 and by a gift to Georgetown University.

Uri Stemmer was partially supported by the Israel Science Foundation (grant 1871/19) and by Len Blavatnik and the Blavatnik Family foundation.

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

## A   General Variant of challenge-DP

**Definition A.1.** *Consider an algorithm $\mathcal{M}$ that, in each phase $i \in [T]$, conducts an arbitrary interaction with the $i$th user. We say that algorithm $\mathcal{M}$ is $(\varepsilon, \delta)$-challenge differentially private if for any adversary $\mathcal{B}$ we have that $\texttt{GeneralGame}_{\mathcal{M},\mathcal{B},T}$, defined below, is $(\varepsilon, \delta)$-differentially private (w.r.t. its input bit $b$).*

---

**Algorithm 8** $\texttt{GeneralGame}_{\mathcal{M},\mathcal{B},T}$

---

**Setting:** $T \in \mathbb{N}$ denotes the number of phases. $\mathcal{M}$ is an interactive algorithm and $\mathcal{B}$ is an adaptive and interactive adversary.

**Input of the game:** A bit $b \in \{0, 1\}$. (The bit $b$ is unknown to $\mathcal{M}$ and $\mathcal{B}$.)

1. For $i = 1, 2, \ldots, T$
   (a) The adversary $\mathcal{B}$ outputs a bit $c_i \in \{0, 1\}$, under the restriction that $\sum_{j=1}^{i} c_j \leq 1$.
   (b) The adversary $\mathcal{B}$ chooses two interactive algorithms $\mathcal{I}_{i,0}$ and $\mathcal{I}_{i,1}$, under the restriction that if $c_i = 0$ then $\mathcal{I}_{i,0} = \mathcal{I}_{i,1}$.
   (c) Algorithm $\mathcal{M}$ interacts with $\mathcal{I}_{i,b}$. Let $\hat{y}_i$ denote the view of $\mathcal{I}_{i,b}$ at the end of this interaction.
   (d) If $c_i = 0$ then set $\tilde{y}_i = \hat{y}_i$. Otherwise set $\tilde{y}_i = \perp$.
   (e) The adversary $\mathcal{B}$ obtains $\tilde{y}_i$.
2. Output $\mathcal{B}$'s view of the game.

---

# B  Group privacy

In this section we prove Theorem 3.4.

*Proof of Theorem 3.4.* Fix $g \in \mathbb{N}$ and fix an adversary $\mathcal{B}$ (that poses at most $g$ challenge rounds). We consider a sequence of games $\mathcal{W}_0, \mathcal{W}_1, \ldots, \mathcal{W}_g$, where $\mathcal{W}_\ell$ is defined as follows.

1. Initialize algorithm $\mathcal{M}$ and the adversary $\mathcal{B}$.
2. For round $i = 1, 2, \ldots, T$:
   (a) Obtain a challenge indicator $c_i$ and two labeled inputs $(x_{i,0}, y_{i,0})$ and $(x_{i,1}, y_{i,1})$ from $\mathcal{B}$.
   (b) If $\sum_{j=1}^{i} c_j > \ell$ then set $(w_i, z_i) = (x_{i,0}, y_{i,0})$. Otherwise set $(w_i, z_i) = (x_{i,1}, y_{i,1})$.
   (c) Feed $w_i$ to algorithm $\mathcal{M}$, obtain an outcome $\hat{y}_i$, and feed it $z_i$.
   (d) If $c_i = 0$ then set $\tilde{y}_i = \hat{y}_i$. Otherwise set $\tilde{y}_i = \perp$.
   (e) Give $\tilde{y}_i$ to $\mathcal{B}$.
3. Output $\tilde{y}_1, \ldots, \tilde{y}_T$ and the internal randomness of $\mathcal{B}$.

That is, $\mathcal{W}_\ell$ simulates the online game between $\mathcal{M}$ and $\mathcal{B}$, where during the first $\ell$ challenge rounds algorithm $\mathcal{M}$ is given $(x_{i,1}, y_{i,1})$, and in the rest of the challenge rounds algorithm $\mathcal{M}$ is given $(x_{i,0}, y_{i,0})$. Note that

$$\texttt{OnlineGame}_{\mathcal{M},\mathcal{B},T,g}(0) \equiv \mathcal{W}_0 \qquad \text{and} \qquad \texttt{OnlineGame}_{\mathcal{M},\mathcal{B},T,g}(1) \equiv \mathcal{W}_g.$$

We claim that for every $0 < \ell \leq g$ it holds that $\mathcal{W}_{\ell-1} \approx_{(\varepsilon,\delta)} \mathcal{W}_\ell$. To this end, fix $0 < \ell \leq g$ and consider an adversary $\widehat{\mathcal{B}}$, that poses at most one challenge, defined as follows. Algorithm $\widehat{\mathcal{B}}$ runs $\mathcal{B}$ internally. In every round $i$, algorithm $\widehat{\mathcal{B}}$ obtains from $\mathcal{B}$ a challenge bit $c_i$ and two labeled inputs $(x_{i,0}, y_{i,0})$ and $(x_{i,1}, y_{i,1})$. As long as $\mathcal{B}$ did not pose its $\ell$th challenge, algorithm $\widehat{\mathcal{B}}$ outputs $(x_{i,1}, y_{i,1}), (x_{i,1}, y_{i,1})$. During the round $i$ in which $\mathcal{B}$ poses its $\ell$th challenge, algorithm $\mathcal{B}$ outputs $(x_{i,0}, y_{i,0}), (x_{i,1}, y_{i,1})$. This is the challenge round posed by algorithm $\widehat{\mathcal{B}}$. In every round $t$ afterwards, algorithm $\widehat{\mathcal{B}}$ outputs $(x_{i,0}, y_{i,0}), (x_{i,0}, y_{i,0})$. When algorithm $\widehat{\mathcal{B}}$ obtains an answer $\tilde{y}_i$ it sets $\tilde{\tilde{y}}_i = \begin{cases} \tilde{y}_i, & \text{if } c_i = 0 \\ \perp, & \text{if } c_i = 1 \end{cases}$ and gives $\tilde{\tilde{y}}_i$ to algorithm $\mathcal{B}$.

As $\widehat{\mathcal{B}}$ is an adversary that poses (at most) one challenge, by the privacy properties of $\mathcal{M}$ we know that $\texttt{OnlineGame}_{\mathcal{M},\widehat{\mathcal{B}},T}$ is $(\varepsilon, \delta)$-DP. Recall that the output of $\texttt{OnlineGame}_{\mathcal{M},\widehat{\mathcal{B}},T}$ includes all of the randomness of $\widehat{\mathcal{B}}$, as well as the answers $\tilde{y}_t$ generated throughout the game. This includes the randomness of $\mathcal{B}$ (which $\widehat{\mathcal{B}}$ runs internally), and hence, determines also all of the $\tilde{\tilde{y}}_i$'s defined by $\widehat{\mathcal{B}}$ throughout the interaction. Let $P$ be a post-processing procedure that takes the output of $\texttt{OnlineGame}_{\mathcal{M},\widehat{\mathcal{B}},T}$ and returns the randomness of $\mathcal{B}$ as well as $(\tilde{\tilde{y}}_1, \ldots, \tilde{\tilde{y}}_T)$. By closure of DP to post-processing, we have that $P(\texttt{OnlineGame}_{\mathcal{M},\widehat{\mathcal{B}},T}(0)) \approx_{(\varepsilon,\delta)} P(\texttt{OnlineGame}_{\mathcal{M},\widehat{\mathcal{B}},T}(1))$. Now note that $P(\texttt{OnlineGame}_{\mathcal{M},\widehat{\mathcal{B}},T}(0)) \equiv \mathcal{W}_{\ell-1}$, and $P(\texttt{OnlineGame}_{\mathcal{M},\widehat{\mathcal{B}},T}(1)) \equiv \mathcal{W}_\ell$, and hence $\mathcal{W}_{\ell-1} \approx_{(\varepsilon,\delta)} \mathcal{W}_\ell$. Overall we have that

$$\texttt{OnlineGame}_{\mathcal{A},\mathcal{B},T,g}(0) \equiv \mathcal{W}_0 \approx_{(\varepsilon,\delta)} \mathcal{W}_1 \approx_{(\varepsilon,\delta)} \mathcal{W}_2 \approx_{(\varepsilon,\delta)} \cdots \approx_{(\varepsilon,\delta)} \mathcal{W}_g \equiv \texttt{OnlineGame}_{\mathcal{A},\mathcal{B},T,g}(1).$$

This shows that $\texttt{OnlineGame}_{\mathcal{A},\mathcal{B},T,g}$ is $(g\varepsilon, g \cdot e^{\varepsilon g} \cdot \delta)$-differentially private, thereby completing the proof. $\qquad \square$

# C  Privacy Analysis of `ChallengeAT`

The privacy guarantees of `ChallengeAT` are captured using the game specified in algorithm `ChallengeAT-Game`$_\mathcal{B}$.

**Theorem C.1.** *For every adversary $\mathcal{B}$ it holds that `ChallengeAT-Game`$_\mathcal{B}$ is $(O(\varepsilon), O(\delta))$-DP w.r.t. the bit $b$ (the input of the game).*

---

**Algorithm 9 ChallengeAT-Game$_\mathcal{B}$**

---

**Setting:** $\mathcal{B}$ is an adversary that adaptively determines the inputs to `ChallengeAT`.
**Input of the game:** A bit $b \in \{0, 1\}$. (The bit $b$ is unknown to `ChallengeAT` and $\mathcal{B}$.)

    1. The adversary $\mathcal{B}$ specifies two neighboring datasets $S_0, S_1 \in X^*$.

    2. Instantiate `ChallengeAT` with the dataset $S_b$ and parameters $\varepsilon, \delta$, threshold $t$, and number of positive reports $r$.

    3. For $i = 1, 2, 3, \ldots$

        (a) Get bit $c_i \in \{0, 1\}$ from $\mathcal{B}$ subject to the restriction that $\sum_{j=1}^{i} c_j \leq 1$.
           % When $c_i = 1$ this is the *Challange round*.

        (b) Get two queries $f_i^0 : X^* \to \mathbb{R}$ and $f_i^1 : X^* \to \mathbb{R}$ from $\mathcal{B}$, each with sensitivity $\Delta$, subject to the restriction that if $c_i = 0$ then $f_i^0 \equiv f_i^1$.

        (c) Give the query $f_i^b$ to `ChallengeAT` and get back the bit $\sigma_i$.

        (d) If $c_i = 0$ then set $\hat{y}_i = \sigma_i$. Otherwise set $\hat{y}_t = \perp$.

        (e) Give $\hat{y}_i$ to the adversary $\mathcal{B}$.

    4. Publish $\mathcal{B}$'s view of the game, that is $\hat{y}_1, \ldots, \hat{y}_T$ and the internal randomness of $\mathcal{B}$.

---

*Proof.* Fix an adversary $\mathcal{B}$. Let `CATG` denote the algorithm `ChallengeAT-Game`$_\mathcal{B}$ with this fixed $\mathcal{B}$. Consider a variant of algorithm `CATG`, which we call `CATG`-noCount defined as follows. During the challenge round $i$, inside the call to `ChallengeAT`, instead of feeding $\sigma_i$ to the `PrivateCounter` we simply feed it 0 (in Step 3d of `ChallengeAT`).

By the privacy properties of `PrivateCounter` (Theorem 2.10), for every $b \in \{0, 1\}$ we have that

$$\texttt{CATG}(b) \approx_{(\varepsilon, 0)} \texttt{CATG-noCount}(b),$$

so it suffices to show that `CATG`-noCount is DP (w.r.t. $b$). Now observe that the execution of `PrivateCounter` during the execution of `CATG`-noCount can be simulated from the view of the adversary $\mathcal{B}$ (the only bit that `ChallengeAT` feeds the counter which is not in the view of the adversary is the one of the challange round which we replaced by zero in `CATG`-noCount). Hence, we can generate the view of $\mathcal{B}$ in algorithm `CATG` by interacting with `AboveThreshold` instead of with `ChallengeAT`. This is captured by algorithm `CAT-G`-AboveThrehold.

---

**Algorithm 10 CATG-AboveThreshold**

---

**Setting:** $\mathcal{B}$ is an adversary that adaptively determines the inputs to `ChallengeAT`.
**Input of the game:** A bit $b \in \{0, 1\}$. (The bit $b$ is unknown to `ChallengeAT` and $\mathcal{B}$.)

    1. The adversary $\mathcal{B}$ specifies two neighboring datasets $S_0, S_1 \in X^*$.

    2. Instantiate `PrivateCounter`

    3. Instantiate `AboveThreshold` on the dataset $S_b$ with parameters $\varepsilon, \delta, t, (r + \lambda)$.

    4. For $i = 1, 2, 3, \ldots$

        (a) Get bit $c_i \in \{0, 1\}$ from the adversary $\mathcal{B}$ subject to the restriction that $\sum_{j=1}^{i} c_j \leq 1$.

        (b) Get two queries $f_i^0 : X^* \to \mathbb{R}$ and $f_i^1 : X^* \to \mathbb{R}$, each with sensitivity $\Delta$ from $\mathcal{B}$, subject to the restriction that if $c_i = 0$ then $f_i^0 \equiv f_i^1$.

        (c) Give the query $f_i^b$ to Algorithm `AboveThreshold` and get back a bit $\sigma_i$.

        (d) If $c_i = 0$ then set $\hat{y}_i = \sigma_i$. Otherwise set $\hat{y}_t = \perp$.

        (e) Give $\hat{y}_i$ to the adversary $\mathcal{B}$.

        (f) If $c_i = 0$ then feed $\sigma_i$ to `PrivateCounter`, and otherwise feed it 0.

        (g) Let $\text{count}_i$ denote the current output of `PrivateCounter`, and HALT if $\text{count}_i \geq r$

    5. Publish $\mathcal{B}$'s view of the game, that is $\hat{y}_1, \ldots, \hat{y}_T$ and the internal randomness of $\mathcal{B}$.

---

This algorithm is almost identical to `CATG`-noCount, except for the fact that `AboveThreshold` might halt the execution itself (even without the halting condition on the outcome of `PrivateCounter`). However, by the utility guarantees of `PrivateCounter`, with probability at least $1 - \delta$ it never errs

by more than $\lambda$, in which case algorithm `AboveThreshold` never halts prematurely. Hence, for every bit $b \in \{0,1\}$ we have that

$$\texttt{CATG-AboveThreshold}(b) \approx_{(0,\delta)} \texttt{CATG-noCount}(b).$$

So it suffices to show that `CATG-AboveThreshold` is DP (w.r.t. its input bit $b$). This almost follows directly from the privacy guarantees of `AboveThreshold`, since `CATG-AboveThreshold` interacts only with this algorithm, except for the fact that during the challenge round $i$ the adversary $\mathcal{B}$ specifies two queries (and only one of them is fed into `AboveThreshold`). To bridge this gap, we consider one more (and final) modification to the algorithm, called $\widehat{\texttt{CATG}}$-AboveThreshold. This algorithm is identical to `CATG-AboveThreshold`, except that in Step 4c we do not feed $f_i^b$ to `AboveThreshold` if $c_i = 1$. That is, during the challenge round we do not interact with `AboveThreshold`.

Now, by the privacy properties of `AboveThreshold` we have that $\widehat{\texttt{CATG}}$-AboveThreshold is DP (w.r.t. its input bit $b$). Furthermore, when algorithm `AboveThreshold` does not halt prematurely, we have that $\widehat{\texttt{CATG}}$-AboveThreshold is identical to `CATG-AboveThreshold`. Therefore, for every bit $b \in \{0,1\}$ we have

$$\texttt{CATG-AboveThreshold}(b) \approx_{(0,\delta)} \widehat{\texttt{CATG}}\text{-AboveThreshold}(b).$$

Overall we get that

$$
\begin{aligned}
\texttt{CATG(0)} &\approx_{(\varepsilon,0)} \texttt{CATG-noCount}(0) \\
&\approx_{(0,\delta)} \texttt{CATG-AboveThreshold}(0) \\
&\approx_{(0,\delta)} \widehat{\texttt{CATG}}\text{-AboveThreshold}(0) \\
&\approx_{(\varepsilon,\delta)} \widehat{\texttt{CATG}}\text{-AboveThreshold}(1) \\
&\approx_{(0,\delta)} \texttt{CATG-AboveThreshold}(1) \\
&\approx_{(0,\delta)} \texttt{CATG-noCount}(1) \\
&\approx_{(\varepsilon,0)} \texttt{CATG(1)}
\end{aligned}
$$

$\square$

# D   Privacy Analysis of `POP`

In this section we prove Theorem 5.2.

*Proof of Theorem 5.2.* Let $\mathcal{B}$ be an adversary that plays in `OnlineGame` against POP, posing at most 1 challenge. That is, at one time step $i$, the adversary specifies two inputs $(x_i^0, y_i^0), (x_i^1, y_i^1)$, algorithm POP processes $(x_i^b, y_i^b)$, and the adversary does not see the prediction $\hat{y}_i$ at this time step. We need to show that the view of the adversary is DP w.r.t. the bit $b$. To show this, we observe that the view of $\mathcal{B}$ can be generated (up to a small statistical distance of $\delta$) by interacting with `ChallengeAT` as in the game `ChallengeAT-Game`. Formally, consider the following adversary $\hat{\mathcal{B}}$ that simulates $\mathcal{B}$ while interacting with `ChallengeAT` instead of POP.

As $\hat{\mathcal{B}}$ only interacts with `ChallengeAT`, its view at the end of the execution (which includes the view of the simulated $\mathcal{B}$) is DP w.r.t. the bit $b$. Furthermore, the view of the simulated $\mathcal{B}$ generated in this process is almost identical to the view of $\mathcal{B}$ had it interacted directly with POP. Specifically, the only possible difference is that the computation of $\hat{y}_i$ in Step 3(e)ii of $\hat{\mathcal{B}}$ might not be well-defined. But this does not happen when `ChallengeAT` maintains correctness, which holds with probability at least $1 - \delta$.

Overall, letting $\texttt{ChallengeAT-Game}_{\hat{\mathcal{B}}|_{\mathcal{B}}}$ denote the view of the simulated $\mathcal{B}$ at the end of the interaction of $\hat{\mathcal{B}}$ with `ChallengeAT`, we have that

$$
\begin{aligned}
\texttt{OnlineGame}_{\texttt{POP},\mathcal{B}}(0) &\approx_{(0,\delta)} \texttt{ChallengeAT-Game}_{\hat{\mathcal{B}}|_{\mathcal{B}}}(0) \\
&\approx_{(\varepsilon,\delta)} \texttt{ChallengeAT-Game}_{\hat{\mathcal{B}}|_{\mathcal{B}}}(1) \\
&\approx_{(0,\delta)} \texttt{OnlineGame}_{\texttt{POP},\mathcal{B}}(1).
\end{aligned}
$$

---

**Algorithm 11** $\hat{\mathcal{B}}$

---

**Setting:** This is an adversary that plays against ChallengeAT in the game ChallengeAT-Game.

1. Specify two datasets $S_0 = \{0\}$ and $S_1 = \{1\}$.

2. Instantiate algorithm $\mathcal{B}$

3. For $i = 1, 2, \ldots, T$

   (a) Obtain a challenge indicator $c_i$ and inputs $x_i^0, x_i^1$ from $\mathcal{B}$ (where $x_i^0 = x_i^1$ if $c_i = 0$).

   (b) Let $\ell_i \in [k]$ be chosen uniformly at random

   (c) Define the query $q_i : \{0, 1\} \to \mathbb{R}$, where $q_i(b) = f_i$ and where $f_i$ is defined as in Step 3e of POP.
   % Note that, given $b$, this can be computed from $(x_1^0, x_1^1), \ldots, (x_i^0, x_i^1)$ and $\ell_1, \ldots, \ell_i$ and $y_1, \ldots, y_{i-1}$. Furthermore, whenever $c_i = 0$ then this is a query of sensitivity at most 1. When $c_i = 1$ the sensitivity might be large, which we view it as *two* separate queries, corresponding to a challenge round when playing against ChallengeAT.

   (d) Output the challenge bit $c_i$ and the query $q_i$, which is given to ChallengeAT.

   (e) If $c_i = 0$ then
       i. Obtain an outcome $\sigma_i$ from ChallengeAT
       ii. Define $\hat{y}_i$ as in Step 3f of POP, as a function of $\sigma_i$ and $(x_1^0, x_1^1), \ldots, (x_i^0, x_i^1)$ and $\ell_1, \ldots, \ell_i$ and $y_1, \ldots, y_{i-1}$.
       iii. Feed the bit $\hat{y}_i$ to the adversary $\mathcal{B}$

   (f) Obtain a "true" label $y_i$ from the adversary $\mathcal{B}$.

---

$\square$

# E    A Coin Flipping Game

Consider algorithm 12 which specifies an $m$-round "coin flipping game" against an adversary $\mathcal{B}$. In this game, the adaptively chooses the biases of the coins we flip. In every flip, the adversary might gain a reward or incur a "budget loss". The adversary aims to maximize the rewards it collects before its budget runs out.

---

**Algorithm 12** $\texttt{CoinGame}_{\mathcal{B},k,m}$

---

**Setting:** $\mathcal{B}$ is an adversary that determins the coin biases adaptively. $k$ denotes the "budget" of the adversary. $m$ denotes the number of iterations.

1. Set $\text{budget} = k$ and $\text{reward} = 0$.

2. In each round $i = 1, 2, \ldots, m$:

   (a) The adversary chooses $0 \leq p_i \leq \frac{5}{6}$ and $\frac{p_i}{5} \leq q_i \leq 1 - p_i$, possibly based on the first $(i-1)$ rounds.

   (b) A random variable $X_i \in \{0, 1, 2\}$ is sampled, where $\Pr[X_i = 1] = p_i$ and $\Pr[X_i = 2] = q_i$ and $\Pr[X_i = 0] = 1 - p_i - q_i$.

   (c) The adversary obtains $X_i$

   (d) If $X_i = 1$ and $\text{budget} > 0$ then $\text{reward} = \text{reward} + 1$.

   (e) Else if $X_i = 2$ then $\text{budget} = \text{budget} - 1$.

3. Output $\text{reward}$.

---

The next theorem states that no adversary can obtain reward much larger than $k$ in this game. Intuitively, this holds because in every time step $i$, the probability of $X_i = 2$ is not much smaller than the probability that $X_i$, then (w.h.p.) it is very unlikely that the number of rewards would be much larger than $k$.

**Theorem E.1** ([Gupta et al., 2010, Kaplan et al., 2021]). *For every adversary's strategy, every $k \geq 0$, every $m \in \mathbb{N}$, and every $\lambda \in \mathbb{R}$, we have*

$$\Pr[\texttt{CoinGame}_{\mathcal{B},k,m} > \lambda] \leq \exp\left(-\frac{\lambda}{6} + 3(k+1)\right).$$

# F Extension to the Agnostic Case

In this section we extend the analysis of POP to the agnostic setting. We use the tilde-notation to hide logarithmic factors in $T, \frac{1}{\delta}, \frac{1}{\beta}, \frac{1}{\varepsilon}$.

**Theorem F.1** ([Ben-David et al., 2009]). *For any hypothesis class $H$ and scalar $M^* \geq 0$ there exists an online learning algorithm such that for any sequence $((x_1, y_1), \ldots, (x_T, y_T))$ satisfying $\min_{h \in H} \sum_{i=1}^{T} |h(x_i) - y_i| \leq M^*$ the predictions $\hat{y}_1, \ldots, \hat{y}_T$ given by the algorithm satisfy*

$$\sum_{i=1}^{T} |\hat{y}_i - y_i| \leq O\left(M^* + \text{Ldim}(H) \ln(T)\right).$$

**Definition F.2.** *For parameters $u < w$, let $\text{POP}_{[u,w]}$ denote a variant of POP in which we halt the execution after the $v$th time in which we err, for some arbitrary value $u \leq v \leq w$. (Note that the execution might halt even before that, by the halting condition of POP itself.) This could be done while preserving privacy (for appropriate values of $u < w$) by using the counter of Theorem 2.10 for privately counting the number of mistakes.*

**Lemma F.3.** *Let $H$ be a hypothesis class with $d = \text{Ldim}(H)$, and let $\mathcal{A}$ denote the non-private algorithm from Theorem F.1 with $M^* = d \ln(T)$. Denote $k = \tilde{\Theta}\left(\frac{d^2}{\varepsilon}\right)$, $r = u = \Theta\left(kd \ln(T)\right)$, and $w = 2u$. Consider executing $\text{POP}_{[u,w]}$ with $\mathcal{A}$ and with parameters $k, r$ on an adaptively chosen sequence of inputs $(x_1, y_1), \ldots, (x_{i^*}, y_{i^*})$, where $i^* \leq T$ denotes the time at which $\text{POP}_{[u,w]}$ halts. Then, with probability at least $(1 - \beta)$ it holds that*

$$\text{OPT}_{i^*} \triangleq \min_{h \in H} \sum_{i=1}^{i^*} |h(x_i) - y_i| > d \cdot \ln(T).$$

*Proof sketch.* Similarly to the proof of Theorem 5.3, we set $k = \tilde{\Omega}\left(\frac{d^2}{\varepsilon}\right)$, and assume that if less than $\frac{1}{5}$ the experts disagree with the other experts, then algorithm $\text{POP}_{[u,w]}$ returns the majority vote with probability 1.

Let $1/5$-Err denote the random variable that counts the number of time steps in which at least $1/5$th of the experts make an error. As in the proof of Theorem 5.3, $1/5$-Err upper bounds both the number of mistakes made by $\text{POP}_{[u,w]}$, which we denote by $\text{OurError}$, as well as the number of times in which algorithm $\texttt{ChallengeAT}$ returns an "above threshold" answer, which we denote by $\text{NumTop}$. By Theorem 4.2, we know that (w.h.p.) $\text{NumTop} \geq r$. Also let $\text{WorstExpert}$ denote the largest number of mistakes made by a single expert.

Consider the time $i^*$ at which $\text{POP}_{[u,w]}$ halts. If it halts because $u \leq v \leq w$ mistakes have been made, then

$$k \cdot \text{WorstExpert} \geq 1/5\text{-Err} \geq \text{OurError} \geq u = \Omega\left(kd \ln(T)\right).$$

Alternatively, if $\text{POP}_{[u,w]}$ halts after $r$ "above threshold" answer, then

$$k \cdot \text{WorstExpert} \geq 1/5\text{-Err} \geq \text{NumTop} \geq r = \Omega\left(kd \ln(T)\right).$$

At any case, when $\text{POP}_{[u,w]}$ halts it holds that at least one expert made at least $\Omega\left(d \ln(T)\right)$ mistakes. Therefore, by Theorem F.1, we have that $\text{OPT}_{i^*} \geq d \ln(T)$.

$\square$

**Theorem F.4.** *Let $H$ be a hypothesis class with $\text{Ldim}(H) = d$. There exists an $(\varepsilon, \delta)$-Challenge-DP online learning algorithm providing the following guarantee. When executed on an adaptively*

*chosen sequence of inputs* $(x_1, y_1), \ldots, (x_T, y_T)$, *then the algorithm makes at most* $\tilde{O}\left(\frac{d \cdot \text{OPT}}{\varepsilon^2} + \frac{d^2}{\varepsilon^2}\right)$ *mistakes (w.h.p.), where*

$$\text{OPT} \triangleq \min_{h \in H} \sum_{i=1}^{T} |h(x_i) - y_i|.$$

*Proof sketch.* This is obtained by repeatedly re-running $\text{POP}_{[u,w]}$, with the parameter setting specified in Lemma F.3. We refer to the time span of every single execution of $\text{POP}_{[u,w]}$ as a *phase*.

By construction, in every phase, $\text{POP}_{[u,w]}$ makes at most $w = \tilde{\Theta}(kd)$ mistakes. By Lemma F.3 *every* hypothesis in $H$ makes at least $d \cdot \ln(T)$ mistakes in this phase. Therefore, there could be at most $\tilde{O}\left(\max\left\{1, \frac{\text{OPT}}{d}\right\}\right)$ phases, during which we incur a total of at most $\tilde{O}\left(\frac{d \cdot \text{OPT}}{\varepsilon^2} + \frac{d^2}{\varepsilon^2}\right)$ mistakes. $\qquad\square$

