# OpenReview forum: "Black-Box Differential Privacy for Interactive ML"
_NeurIPS.cc/2023/Conference — NeurIPS 2023 poster_

### Official Review · Reviewer_4d8Q · 2023-06-23

**Soundness:** 3 good
**Presentation:** 1 poor
**Contribution:** 2 fair
**Rating:** 3
**Confidence:** 4

**Summary:**

This work proposes a novel method to apply DP to the context of interactive ML.

**Strengths:**

The method itself is an interesting proposal, well-supported with proofs and theory. This work also extensively describes prior advances in the field in great detail.

**Weaknesses:**

The paper has numerous issues with the way it is written. Firstly, there should not be any citations in the abstract, so, please reword it. Secondly, and much more importantly, this paper lacks a proper conclusion. There is no way for the reader to get a clear overview of the implications of this work as well as its relevance to the broader ML community. Instead, the work ends with a large algorithm definition, which really either belongs to the methods section or to the appendix. While I understand the overall idea of what the work proposes, there are no clear contributions either. I strongly encourage authors to rework these, as otherwise it is incredibly difficult to identify the merits of their method.

Certain sections really don’t belong where they are right now: 1.1 in my eyes has nothing to do with the introduction; example 2.6, while helps contextualise the method is really taking the space away from the rest of the work; many algorithms can be placed in the appendix etc. Point being that the work is very difficult to process because some crucial sections are missing, yet some sections which were included really do not add the value to the submission.

Finally, it looks to me as this is a purely theoretical paper, there are no experiments to support what the authors proposed. Given that the submission is presented as application-driven and useful for ML practitioners, this is rather odd.


**Questions:**

Are there any practical results to showcase the advantages of the proposed method?

**Limitations:**

Overall, I do not see many reasons to accept this work in its current state: While the method looks promising and may have broad implications for the use of DP in interactive setting, the manuscript does not do the method any justice and really struggles to make this accessible to a wide range of ML researchers.

---

> ### Author Rebuttal · Authors · 2023-08-07
>
> **> there should not be any citations in the abstract**
>
> This is a minor issue, and we are willing to remove the citations
>
>
> **> this paper lacks a proper conclusion; there are no clear contributions; it is incredibly difficult to identify the merits; the manuscript does not do the method any justice**
>
> We have two main contributions:
> 1. We present a meaningful and new privacy model for online classification.
> 2. We present a new algorithm satisfying our privacy notion, that exhibits a doubly exponential improvement in the error compared to prior works on private online classification.
>
> We believe that these two contributions constitute a significant story, and will make this clearer in the paper.
>
>
> **> Certain sections really don't belong where they are right now**
>
> While our algorithms are relatively simple, the proofs are non-trivial and could not fit the main body at its current form. We are open to reorganizing the paper, and will do our best (within the page limit) towards providing more proof details and insights.
>
>
> **> this is a purely theoretical paper, there are no experiments... Are there any practical results?**
>
> This is a theory paper. We agree that concrete bounds matter, but asymptotic bounds are not less important, and the asymptotic improvements we obtain are huge (polynomial overhead vs double exponential). We believe that our work would lead to future studies on this topic, both theoretically and practically oriented.

---

> > ### Comment · Reviewer_4d8Q · 2023-08-14
> > **Response to the rebuttal**
> >
> > I would like to thank the authors for their response to my comments.
> >
> > I am still, however, not convinced that A) the paper is fir for acceptance in its current state (partially because it is missing crucial components such as clear conclusion, contributions etc.) and B) it is possible to address the reviewer comments in time for the deadline. Therefore my score remains unchanged.

---

> > > ### Author Response · Authors · 2023-08-15
> > >
> > > Thank you for your comment.
> > >
> > > **> missing crucial components such as clear conclusion, contributions**
> > >
> > > As we mentioned in the rebuttal, our work presents a meaningful and new privacy model for online classification, together with a new algorithm that exhibits a *doubly exponential improvement* in the error compared to prior works on this topic. We would appreciate any specific comments or feedback as to why these contributions are not "clear".

---

### Official Review · Reviewer_4WCj · 2023-06-27

**Soundness:** 4 excellent
**Presentation:** 3 good
**Contribution:** 4 excellent
**Rating:** 7
**Confidence:** 3

**Summary:**

The paper addresses the problem of privacy preserving interactive learning. In this problem, a recommendation algorithm improves its model by answering private queries performed by a set of parties in sequential rounds. The responses to users queries should adapt to the query made by each party. Therefore, in order to provide an accurate service, the algorithm's responses must be strongly sensitive to the users.
This imposes an important restriction on the privacy guarantees that can be provided using the classical notion of differential privacy (DP): classical DP requires that all responses to queries are equally privacy preserving. Therefore, if the algorithm satisfies this notion of privacy, its accuracy is limited.

To tackle this problem, the contribution proposes challenge DP: a relaxed notion of DP in which the mechanism is allowed to provide a sensitive output to the party that generated the query, while still providing a less sensitive view to other parties. Using challenge DP, the contribution proposes a construction that, only by having black-box access to a interactive learning algorithm, can construct
a privacy preserving algorithm. The construction improves the accuracy of previous work, reducing the number of errors from exponential to quadratic in d, where d is the number of mistakes made by the black-box algorithm. Furthermore, these improvements are achieved while assuming an adversary that participates in the learning process controlling multiple parties and that can adapt its input in one round of the protocol.

**Strengths:**

The paper is very well written and key aspects of the contribution are presented clearly. The contribution is novel: the relaxation of classical DP and the POP construction are clever. The results are significant, as improvements with respect previous work are important while
assumptions on the adversary remain realistic. The overall quality is good.

**Weaknesses:**

I only have fairly minor remarks with respect to the presentation of the contribution. While the introduction to the problem is very nicely achieved, the space dedicated to the main contributions is a bit short. I would have appreciated if it had been bigger and had provided more details. Also, there are multiple DP definitions (such as Definitions 2.1, 2.7 and 2.9) that overload the statement "X satisfies (epsilon, delta)-DP" and then understanding which algorithm satisfies which definition of DP becomes a bit confusing. Different terminology for each case could help in clarity.

**Questions:**

1- In the definitions of DP under adaptive queries (Def 2.7) and adaptive inputs (Def 2.9), it seems that the neighboring databases are
always defined with respect to the bit b. As I understand, this means that the database only changes on the query in which the adversary
plans to be adaptive. I have the impression that this might not reflect some realistic scenarios in interactive ML. For example, consider the case in which the adversary controls all the parties except of one party: Alice. It would be nice that the output (i.e. the view of the adversary) is differentially private with respect to the participation of Alice or not (which, in my understanding, is independent of bit b). Do the aforementioned definitions contemplate this case?

2- In Theorems 4.1 and 4.2, you show how the POP construction (Algorithm 7) performs when it assumes an adversary that can adapt in a single round. However, you define challenge DP to be compatible with many adaptive/challenge rounds (group privacy). How an increase in the number of challenge rounds allowed to the adversary (i.e. an increase of g) would impact in the privacy/accuracy trade-offs?

**Limitations:**

While theoretical improvements are significant, more experimentation is required to understand if the practical deployment of the construction is feasible.

---

> ### Author Rebuttal · Authors · 2023-08-07
>
> **> The space dedicated to the main contributions is a bit short**
>
> While our algorithms are relatively simple, the proofs are non-trivial and could not fit the main body at its current form. We are open to reorganizing the paper, and will do our best (within the page limit) towards providing more proof details and insights.
>
>
> **> In definitions 2.7 and 2.9 ... it seems that the neighboring databases are always defined with respect to the bit b. As I understand, this means that the database only changes on the query in which the adversary plans to be adaptive**
>
> The adversary is adaptive throughout the execution: The bit b indexes two thought experiments which the adversary is trying to distinguish between. At every step of the execution, the adversary adaptively determines an input to be included in the execution, except for Alice’s input where the adversary specifies *two* possible inputs and only one of them is included in the execution (based on the value of the bit b which is unknown to the adversary). The adversary's goal is to distinguish between the executions with b=0 and b=1. In other words, the adversary tries to figure out which of Alice's inputs was included in the execution.
>
> Adaptivity on every step makes a strong adversary. In particular, it might adaptively choose future inputs in order to try and guess the bit b (i.e., try to guess Alice's input).
>
>
> **> Consider an adversary controlling all the parties except for Alice. It would be nice if the view of the adversary is DP with respect to the participation of Alice or not**
>
> As is standard in the literature, these privacy notions could be stated in two flavors: Either we protect against an arbitrary *change* to one individual's data (say Alice), or we protect against the *addition/removal* of one individual. These variants also exist w.r.t. the standard definition of DP, and are very similar. Definitions 2.7 and 2.9 are stated w.r.t. the first option, but they could easily be modified for the second option (without any real effect on the rest of the paper).
>
>
> **> In Theorems 4.1 and 4.2... how an increase in the number of challenge rounds allowed to the adversary (i.e. an increase of g) would impact the privacy/accuracy trade-offs?**
>
> This could be obtained from our group-privacy theorem (Theorem 3.4). Specifically, Theorem 3.4 shows that POP (as is, with the same utility guarantees) satisfies $(g\cdot\varepsilon , g\cdot e^{\varepsilon g}\cdot \delta)$-Challenge-DP. Alternatively, by rescaling $\varepsilon$ and $\delta$, you could get an $(\varepsilon,\delta)$-Challenge-DP algorithm whose utility guarantees are degraded by a roughly $g^2$ factor (compared to Theorem 4.2).

---

> > ### Comment · Reviewer_4WCj · 2023-08-16
> >
> > I thank the authors for their responses. Now my doubts are clarified. While the presentation could be improved by restructuring the content of the main text, I think the work presents a strong contribution that importantly reduces the overhead with respect to previous solutions. Therefore my score is likely to remain unchanged.

---

### Official Review · Reviewer_FnVa · 2023-07-08

**Soundness:** 2 fair
**Presentation:** 3 good
**Contribution:** 2 fair
**Rating:** 6
**Confidence:** 3

**Summary:**

This paper studies privacy in the setting of interactive machine learning processes.  Challenge DP is presented as a new relaxation of DP that is satisfies many of the desirable properties of DP and any non-private online prediction algorithm can be constructed into a Challenge DP online prediction algorithm.


**Strengths:**

The paper tackles an important problem and provides a nice motivating example in the setting of a continually improving chatbot.  It is clear from the example that the interaction is adaptive.  Rather than requiring that the interaction with the chatbot be differentially private, they study a relaxation called joint differential privacy, allowing the transcript of responses to depend arbitrarily on each individual’s prompts, but the chatbot should not leak too much information about others’ conversations in each response.  There has been work for this problem in the setting of (traditional) DP, but this is the first work to study mistake bounds in the setting of Joint DP, or more specifically Challenge DP (interactive variant of JDP), which they define and prove basic properties of.

**Weaknesses:**

The motivating example is with a chatbot, but the problem setting is for private online classification, which might not be a relevant problem for a chatbot, as it is not clear what the labels or mistakes of a chatbot would be.  Is there a motivating example that is more related to private online classification?

The only proof in the paper is proving group privacy of challenge DP.  Although group privacy is an important property of any privacy definition, I do not think it adds to the overall narrative of the paper and not surprising as challenge DP is still related to DP.  The main results of this paper are in Theorems 4.1 and 4.2, so the paper should highlight the analysis there, or at the least provide a proof sketch. From the introduction, it is not clear why the reader should care about a group privacy property.   Furthermore, designing challenge DP algorithms is not clear and its relation with creating an online game that is DP is not clear (as in Theorem 4.1).

Although the problem is well motivated, the full story still seems incomplete. In particular, is there an actual gap between what is achievable under joint DP and (traditional) DP?  Currently, there is an existing DP approach that achieves very large mistake bound and this paper achieves a much better mistake bound, but is there a lower bound result?

Is there just a typo in Algorithm 1 and 6 where noise is not added to the threshold?  The privacy analysis of AboveThreshold depends on noise being added to the threshold that is reused at each round that is not above the noisy threshold.  I am willing to increase my score if this is merely a typo but would like the authors to verify.


Minor:
- \mathcal{M} denoted mistake bounds and then in Definition 2.5 \mathcal{M} is a mechanism.

- Line 116 “if the algorithm errs than at least”

- What is the optimal JDP approach for example 2.6?  Seems like you need to know the median to know which ones are above or below.

- Footnote 1 on page 2, “as =composition”.

- Line 309 “That is, For every”

[Update] I have increased my score based on the rebuttal.

**Questions:**

is there an actual gap between what is achievable under joint (challenge) DP and (traditional) DP?

Is there just a typo in Algorithm 1 and 6 where noise is not added to the threshold?

**Limitations:**

The problem setting is for private online classification, yet the relation to a chatbot example is not quite clear.

---

> ### Author Rebuttal · Authors · 2023-08-07
>
> **> The motivating example is with a chatbot... is there an example that is more related to private online classification?**
>
> 1. A hospital conducting a study on a new disease might use private online classification to predict the risk of an individual having this disease (based on available tests and medical history).
>
> 2. A bank might use private online classification to decide whether or not to grant an individual a loan.
>
> 3. An online seller might use private online classification to decide whether to suggest a promotion or not to a user.
>
>
> **> The only proof in the paper is proving group privacy of challenge DP**
>
> While our algorithms are relatively simple, the proofs are non-trivial and could not fit the main body at its current form. We are open to reorganizing the paper, and will do our best (within the page limit) towards providing more proof details and insights.
>
>
> **> the full story still seems incomplete... is there an actual gap?**
>
> We are not aware of a lower bound / separation result. In our work, we present a meaningful and new privacy model for online classification, together with a new algorithm that exhibits a doubly exponential improvement in the error compared to prior works on this topic. We believe that this tells a significant story, even without a separation result.
>
>
> **> typo in Algorithm 1 and 6 where noise is not added to the threshold? I am willing to increase my score if this is merely a typo**
>
> This is not a typo. It is true that the standard presentation of AboveThreshold utilizes a noisy threshold, which allows it to satisfy pure (eps,0)-DP. But the algorithm remains private even without the noise on the threshold, in which case it satisfies approx (eps,delta)-DP. This appears, for example, in [Hardt and Rothblum, 2010] and [Kaplan et al., 2021] (which we cited).
>
> In the standard formulation of AboveThreshold, we add noise to the threshold, and resample this noise after every "above" answer. This is OK in the standard setting where the queries are considered to be non-private (and only the data points are private). However, as we mentioned in the beginning of Section 4, in our variant (ChallengeAT), the queries are also considered to be private information. Therefore, resampling the noise is problematic as it depends on the current query in a non-private way. (Replacing a constant 0 query with a constant 1 query would typically trigger a new noise sampling.) In contrast, we needed to ensure that the rest of execution behaves similarly no matter what the current query is, and we don't want one query to generate a new noise sampling while the other do not.
>
> We would be grateful if you could increase your score, as you suggested.
>
> (We remark that there are variants of AboveThreshold in which we add noise to the threshold in the beginning of the execution, and never re-sample it, even after an "above" answer. This would be applicable in our setting, but the "noiseless" version helps to slightly simplify our analysis.)

---

> > ### Comment · Reviewer_FnVa · 2023-08-17
> >
> > I have read the rebuttal.  Thanks for answering my questions and for clarifying the lack of noise added to the threshold.  I was not aware of the variant that did not add noise to the threshold.  As promised I will increase my score.  I do feel that the story is still incomplete without addressing whether there is a gap or not.

---

> > > ### Author Response · Authors · 2023-08-17
> > >
> > > Thank you for your comment and for updating the score! We truly believe that challenge DP is a meaningful privacy notion in our setting (and it seems that all of the reviewers agree with us on that point). The fact that we were able to leverage it in order to get a **doubly exponential improvement** is, in our opinion, a very interesting story (even if it raises open questions, which we hope will be addressed by future work).

---

### Official Review · Reviewer_LQAr · 2023-07-12

**Soundness:** 3 good
**Presentation:** 2 fair
**Contribution:** 2 fair
**Rating:** 5
**Confidence:** 3

**Summary:**

The authors propose a new differential privacy definition with desirable online learning properties. In this new variant named Challenge Differential Privacy, the adversary can observe the output of a sequence of online queries, except a "challenge query". In this query, two possible pairs of inputs are picked by an adversary. Based on the transcript of the queries without this challenge step, the adversary needs to decide which data point has been used for the hidden query. The authors introduce an algorithm that satisfies Challenge-DP, in which an ensemble of expert models is used to decide on a majority answer, but only one of the experts is allowed to use the ground truth label of the given sample.


**Strengths:**

* The setting proposed by the authors is interesting. Indeed, the standard definition of Differential Privacy is not well-tailored for online learning.
* The authors discuss and prove the fundamental properties of Differential Privacy for their proposed variant: post-processing, composition, and group privacy.
* The authors propose a technique to achieve ChallengeDP, POP, for which they introduce privacy and utility guarantees.
* The work is compositional; it does not introduce new learning algorithms, but it introduces a technique to convert an existing learning algorithm to a private one that satisfies ChallengeDP.


**Weaknesses:**

* The current work is heavily reliant on the appendix. Consider making the main statements self-contained (ChallengeDP, POP).
* The paper ends with a theorem statement. There is no conclusion/future work/discussion, giving the feel of possibly unfinished work.
* $\overset{\sim}{O}$ notation inside $O$ notation (theorem 4.4) makes it very hard for the reader to compare it to other work/really understand at a granular level the privacy guarantees. As other results have previously shown, constants/linear terms matter a lot in differential privacy; please consider adding explicit guarantees (maybe in the appendix); otherwise, it is hard to understand the benefits of using this approach.
* I would strongly suggest the authors state more clearly that in this online setting, the training algorithm that sees a sample uses it as a training point afterwards (possibly a more explicit protocol like the one in Naor et al. [0], figure 1 could be used).
* A clearer discussion about the difference between private everlasting predictions from Naor et al. [0] and challenge differential privacy could be useful, and why it is a generalization of their work (as stated in the abstract). I acknowledge that the authors provided remark 3.2, but this could be more in-detail justified and discussed, as it is one of the paper's key contributions.

[0]: https://arxiv.org/abs/2305.09579



**Questions:**

* The authors provided concise and clear proof for composition in the interactive setting, but I assume that this setting satisfies even the generalization of composition, namely concurrent composition[0], which is even more suitable for the described problem proposed by the authors. Did the authors consider proving/looking into concurrent composition, as it seems that is a definition well suited for this setting
* In this setting, as far as I understand, the release of the learning algorithms results in a total loss of privacy, is that true, or could it be bounded/there might be a possibility of making a model release in this setting (considering that the initial learning algorithms, before the stream of queries, is not private).
* Do the authors believe that this definition of privacy could replace $(\epsilon, \delta)$-DP for machine learning tasks, or should it be an alternative?
* Is there an equivalence/relationship that the authors observed between $(\epsilon, \delta)$-DP and ChallengeDP?

[0]: https://arxiv.org/abs/2207.08335


**Limitations:**

* Their proposed technique involves maintaining multiple copies of the same model, which is a possibly prohibitive approach when solving the proposed problem in the introduction, given the significant size of expert models.
* The paper ends with a theorem statement. There is no conclusion, further work or discussion. I am willing to increase the score of the paper if the presentation at the end is improved.

---

> ### Author Rebuttal · Authors · 2023-08-07
>
> **> heavily reliant on the appendix**
>
> While our algorithms are relatively simple, the proofs are non-trivial and could not fit the main body at its current form. We are open to reorganizing the paper, and will do our best (within the page limit) towards providing more proof details and insights.
>
>
> **> no conclusion/future work/discussion**
>
> We will add a conclusion section. The short summary is that our work presents a meaningful and new privacy model for online classification, together with a new algorithm that exhibits a doubly exponential improvement in the error compared to prior works on this topic.
>
>
> **> constants/linear terms matter a lot; consider adding explicit guarantees**
>
> We agree that concrete bounds matter, but asymptotic bounds obtained in theoretical work are not less important. The asymptotic improvements we obtain are huge (polynomial vs double exponential overhead). We believe that our work would lead to future studies on this topic, both theoretically and practically oriented.
>
>
> **> clearer discussion about the difference from Naor et al**
>
> Naor et al. considers a setting where the algorithm initially gets a dataset S containing labeled examples from n users. Then, on every time step, a new user arrives and submits its unlabeled example to the algorithm (and the algorithm responds with a predicted label).
>
> We generalize their definition to our setting. Specifically, we extend the definition to capture settings in which every user *interacts* with the algorithm (rather than just submitting its input). In the concrete application we consider (online learning) this corresponds to the user submitting its input, then obtaining the predicted label, and then submitting the "true" label. Our generalized definition (Section A in the supplementary) captures this as a special case and allows for arbitrary interactions with each user. We will make this clearer in Remark 3.2.
>
>
> **> Did the authors consider proving/looking into concurrent composition**
>
> We did not. We believe that the notion does indeed satisfy concurrent composition. Though this seems to require slightly generalizing the existing concurrent composition theorems, because (as they are stated) they assume that the dataset(s) are fixed in the beginning of the execution. In our case the dataset is "evolving".
>
>
> **> the release of the learning algorithms results in a total loss of privacy, is that true?**
>
> Correct. This "relaxation" is what allowed us to improve the bounds given in prior work (by a doubly exponential factor).
>
>
> **> Do the authors believe that this definition could replace DP or should it be an alternative?**
>
> We view it as an alternative that, as we show, would allow you to get better utility in some cases.
>
>
> **> Is there an equivalence/relationship that the authors observed between DP and ChallengeDP?**
>
> The models are not equivalent. The work of Naor et al. showed that PAC learning is possible with ChallengeDP for any concept class with finite VC dimension, which is known not to be the case with the standard notion of DP.

---

> > ### Comment · Reviewer_LQAr · 2023-08-14
> >
> > I appreciate the responses, and I am also looking forward to the discussions with the other reviewers. Overall I consider this work interesting, but there are plenty of improvements to be done on the presentation side.

---

> > > ### Author Response · Authors · 2023-08-15
> > >
> > > Thank you for your comment. As we mentioned in the rebuttal, we will add a conclusion section and we will do our best to reorganize the paper. We would appreciate any specific comments or feedback that could further enhance the clarity.

---

### Decision · Program_Chairs · 2023-09-21

**Decision:**

Accept (poster)

**Comment:**

The reviewers pointed out many strengths of this paper, including the novelty of the problem, the clear need for the definition in machine learning applications, and that the paper provides methods for satisfying the new definition plus analyzes properties of it.

Several reviewers noted that parts of the paper seem incomplete, and provided action items to address this. The authors should address these shortcomings in the final version.

Another major-but-easily-fixable weakness of the paper that the authors should address in the camera ready is the organization of the paper. While the paper does have many strong results, they are mostly in the appendix. The paper contributions start on page 7, with 3.5 pages of intro and 2.5 pages of preliminaries, which only leaves 3 pages for results. This is unusual and seems to diminish the scope of their contribution. The full version of the paper (submitted as supplementary material) contains substantially more technical material and contributions. For the published conference version, the content in the first 6 pages should be substantially reduced to allow for more technical content in the body to better emphasize the contributions.